# Hydrocarbon Degradation and Microbial Survival Improvement in Response to γ-Polyglutamic Acid Application

**DOI:** 10.3390/ijerph192215066

**Published:** 2022-11-16

**Authors:** Ewelina Zając, Monika J. Fabiańska, Elżbieta Jędrszczyk, Tomasz Skalski

**Affiliations:** 1Department of Land Reclamation and Environmental Development, Faculty of Environmental Engineering and Land Surveying, University of Agriculture in Krakow, Al. Mickiewicza 21, 31-120 Krakow, Poland; 2Faculty of Earth Sciences, University of Silesia, 60 Będzińska Street, 41-200 Sosnowiec, Poland; 3Department of Vegetable and Medicinal Plants, Faculty of Biotechnology and Horticulture, University of Agriculture in Krakow, al. 29 Listopada 45, 31-425 Krakow, Poland; 4Tunneling Group, Biotechnology Centre, Silesian University of Technology, Krzywoustego 8, 44-100 Gliwice, Poland

**Keywords:** γ-polyglutamic acid, PGA, petroleum hydrocarbons, remediation, biosurfactant, organic pollutants, hydrocarbon degradation

## Abstract

To improve the environmental sustainability of cleanup activities of contaminated sites there is a need to develop technologies that minimize soil and habitat disturbances. Cleanup technologies, such as bioremediation, are based on biological products and processes, and they are important for the future of our planet. We studied the potential of γ-poly glutamic acid (PGA) as a natural component of biofilm produced by *Bacillus* sp. to be used for the decomposition of petroleum products, such as heavy naphtha (N), lubricating oil (O), and grease (G). The study aimed to assess the impact of the use of different concentrations of PGA on the degradation process of various fractions of petroleum hydrocarbons (PH) and its effect on bacterial population growth in harsh conditions of PH contamination. In laboratory conditions, four treatments of PGA with each of the petroleum products (N, O, and G) were tested: PGA_0_ (reference), PGA_1_ (1% PGA), PGA_1B_ (1% PGA with *Bacillus licheniformis*), and PGA_10_ (10% PGA). After 7, 28, 56, and 112 days of the experiment, the percentage yield extraction, hydrocarbon mass loss, geochemical ratios, pH, electrical conductivity, and microorganisms survival were determined. We observed an increase in PH removal, reflected as a higher amount of extraction yield (growing with time and reaching about 11% in G) and loss of hydrocarbon mass (about 4% in O and G) in all treatments of the PGA compared to the reference. The positive degradation impact was intensive until around day 60. The PH removal stimulation by PGA was also reflected by changes in the values of geochemical ratios, which indicated that the highest rate of degradation was at the initial stage of the process. In general, for the stimulation of PH removal, using a lower (1%) concentration of PGA resulted in better performance than a higher concentration (10%). The PH removal facilitated by PGA is related to the anionic homopoliamid structure of the molecule and its action as a surfactant, which leads to the formation of micelles and the gradual release of PH absorbed in the zeolite carrier. Moreover, the protective properties of PGA against the extinction of bacteria under high concentrations of PH were identified. Generally, the γ-PGA biopolymer helps to degrade the hydrocarbon pollutants and stabilize the environment suitable for microbial degraders development.

## 1. Introduction

Pollutants present in soil and groundwater in Europe are mainly organic pollutants (approx. 59%), heavy metals (approx. 37.3%), and others (approx. 3.6%). In the control group of organic hydrocarbons, the highest percentage is mineral oil (33.7%), but there also exist aromatic hydrocarbons (BETX 6.0%), polycyclic aromatic hydrocarbons (13.3% PAH), phenols (3.6%), and chlorinated hydrocarbons (2.4% CHC) [1]. The main pollutants in European Union (EU) countries are mineral oil and heavy metals, which account for almost 60% of all soil pollutants. The main sources of pollution by petroleum hydrocarbons (PH) are the activities of the petrochemical industry on land and sea, sewage and surface runoff in urban and industrial areas, and spills during the extraction and transport of petroleum [2]. PH enter soils and groundwater, as well as seas and oceans, causing contamination and disrupting the functioning of ecosystems [3,4]. They also pose a toxicological threat to the health and life of living organisms [5,6].

Petroleum products made up from crude oil in the refining process, such as naphtha, oil, and grease, are widely used in transport, as well as in various industries and military areas [7,8]. PH is a group of hundreds of hydrocarbons, differing in structure (alkanes, alkenes, cycloalkanes, aromatic hydrocarbons) and the number of carbon atoms in the molecule (C_6_ ≤ C_35_). The carbon number range is C_6_–C_10_ for gasoline, C_11_–C_28_ for diesel, and C_29_–C_35_ for oil, while lubricants and greases include also high-molecular weight polycyclic aromatic hydrocarbons (PAHs) (4 or more aromatic rings) [9]. From a remediation point of view, PH belong to the group of biodegradable compounds. However, their susceptibility to degradation depends on the complexity of their structure, which is in the order of *n*-alkanes > branched alkanes and alkenes > light *n*-alkyl aromatics > monoaromatics > cyclic alkanes > polycyclic aromatic hydrocarbon > asphaltenes > resins [10].

For remediation of PH contaminated soils, a variety of technologies have been applied [11]. Compared to expensive physical, chemical, and thermal methods, which often have a negative impact on some soil properties and functions [12], technologies based on biological products and processes have the significant potential to be more widely applied. In the natural environment, PH degradation processes take place with the help of microorganisms in the soil. The degradation of crude oil occurs as a result of the use of carbon from hydrocarbon sources for growth and reproduction of microorganisms [13]. The demand for environmentally friendly soil remediation technologies (green remediation) is driving the development of bioremediation research [3,14,15,16] and explains the mechanisms of decomposition of petroleum hydrocarbons by microorganisms [2,13,17,18].

More and more studies focus on supporting the bioremediation process by using various environmentally safe amendments, such as biosurfactants [19,20,21,22,23], or the use of microbial combined methods, including fungi [24,25,26,27], plants [28,29,30,31,32], algae [33], biochar [34,35], and biofilms [36]. γ-polyglutamic acid (PGA) may be one such amendment with great potential for use in the process of pollutant remediation. It is a natural anionic homopoliamide biopolymer synthesized mainly by bacteria of the Gram-positive genus *Bacillus* and *Staphylococus*, as well as fungi (*Saccaromyces cervisiae*) [37,38]. PGA is composed of D- and L-glutamic acid units connected by amide links between a-amino and c-carboxylic acid groups [39]. It is fully biodegradable, non-toxic, environment-friendly, and harmless for humans. The molecular weight of PGA ranges from 7 kDA to 10 kDa, determining the physicochemical properties and biofunctionality of PGA in various fields, especially in agriculture, medicine, wastewater treatment, food, or cosmetics, see [40]. The weight of PGA fractions mostly determines their applications, including good bioflocculant properties (greater than 2000 kDa) [40,41], metal and dye removal (2500–100 kDa) [42], probiotic protectants (300 kDa) [43], and drug delivery tools (50 kDa) [44]. 

In the remediation of pollutants from waters and soils, PGA is mainly used as a flocculant [45,46] for the removal of heavy metals [47,48,49,50], or for dichlorination of chlorinated compounds in groundwater [51]. Research also shows its usefulness in repairing saline-alkali land [52,53]. In the literature, however, there is less research on the possibility of using PGA in the remediation of organic pollutants. However, PGA has been used to good effect as a protective coating for bacteria in the bioremediation process of petroleum contaminated marine sediments [54]. Recently, a positive effect of inoculation of bacterial strains with PGA used for biodegradation of PH in contaminated soils has also been demonstrated [55]. However, there is a lack of information on the effect of PGA alone on various hydrocarbon fractions. The general objective of the study, according to the modern trend of green remediation, is to apply natural biopolymers which are easily biodegradable and have no negative environmental impact. We hypothesize that PGA, as a natural biopolymer and component of biofilm produced by *Bacillus* sp., may accelerate the decomposition of PH and have a protective effect on bacterial populations. We wanted to evaluate the efficiency of different concentrations of PGA on the degradation process of various fractions of PH in controlled laboratory conditions. We also tested the application of PGA as the agent supporting bacterial population growth in harsh conditions of PH contamination. The laboratory experiment is the preliminary phase of research into the application of PGA in the bioremediation of soils contaminated with petroleum substances. Since remediation carried out in natural conditions comprises many complex variables that cannot be controlled, such laboratory simulation is necessary to facilitate future research.

## 2. Materials and Methods

### 2.1. Experiment Set-Up

Research on the influence of PGA on the degradation of various fractions of PH was carried out in laboratory conditions in order to limit the influence of environmental factors on the course of the degradation process. For this reason, PGA in liquid form was applied directly to individual petroleum products (without the addition of soil and water). The laboratory experiment was carried out using five types of treatment with each of the petroleum products, i.e., heavy naphtha (N), lubricating oil (O), and grease (G) in the following combinations:(1)PGA_0_: zeolite + nutrients + petroleum product (reference);(2)PGA_0B_: zeolite + nutrients + petroleum product + bacterial broth;(3)PGA_1_: zeolite + PGA at a 1:100 dilution (1%) + nutrients + petroleum product;(4)PGA_1B_: zeolite + PGA at a 1:100 dilution (1%) with bacterial broth + nutrients + petroleum product;(5)PGA_10_: zeolite + PGA at a 1:10 dilution (10%) + nutrients + petroleum product.

The zeolite, which was used as the carrier, was soaked with either 1% or 10% PGA (except for the reference PGA_0_) in a 10:1 volume ratio. The zeolite was then mixed with the individual petroleum products in a 1:1 volume ratio (50 mL:50 mL) so that PGA was 10% of the petroleum products. In each of the variants, a mixed medium was used in such proportions that the molar ratio of C:N:P was approximately 100:10:1 [56]. As a nutrients source, a commercial mineral fertilizer (NPK) was used. The treatment design was based on results of a previous experiment on the protective role of PGA in the biodegradation of petroleum-contaminated soil [54] and the trial-error principle method [57,58,59].

In the PGA_1B_ treatment, pure cultures of *Bacillus licheniformis* (NCBI1402) at concentration levels of 1 × 10^6^ per 1 mL of broth solution were applied. All ingredients were mixed using a mechanical agitator. 100 mL of the prepared material was placed in each glass container (replicated ten times) and stored in a fume hood, at a constant temperature of 25 °C with a natural flow of air. The sampling times were set at 7, 28, 56, and 112 days (T7–T112). At designated time intervals, samples were weighed and material was collected for chemical and microbiological analysis. Samples for each treatment at each sampling time were prepared in five replicates. 

In the experiment, high concentrations of the petroleum products were used in order to test the effect of PGA in highly contaminated conditions, with particular emphasis on its impact on the survival of the bacterial population.

Because of extinction of bacteria in PGA_0B_ treatment after 7 days, we decided to exclude it from further PH removal analysis. We have only controlled concentration of bacteria in subsequent periods of time to compare it with bacteria concentration in PGA_1B_ treatment.

### 2.2. Media Characteristics

All the petroleum products used in the experiment are commercially available. The following products were used: heavy naphtha (Dragon Poland Sp. z o.o. Sp. k., Skawina, Poland), lubricating oil (Lotos Oil Sp. z o.o., Gdańsk, Poland), and grease (Orlen Oil Sp. z o.o., Kraków, Poland). The chemical composition of all the products is dominated by aliphatic hydrocarbons. *n*-alkanes and branched and cyclic aliphatic hydrocarbons were identified. Heavy naphtha contained compounds in the range of C_5_–C_16_, lubricating oil C_10_–C_27_, and grease C_12_–C_40_.

The PGA from Ambiogel^®^ (Ambioteco Sp. z o.o., Staszów, Poland) contained 5% of pure PGA with various molecular weight (2000–1000 kDa 30%, 1000–140 kDa—50%, 100 kDa–300 Da, 15%, <300 Da—5%). As a carrier, the zeolite (clinoptylite) with particle size 0.2–1.0 mm (Zeocomplex Sp. z o.o., Wieliczka, Poland) was used. To obtain the good conditions for bacterial growth, the NPK fertilizer containing 13.3% total N (5.5% NO_3_, 7.9% NH_4_), 6.1% P_2_O_5_, 17.1% K_2_O and 4.5% MgO, 21.0% SO_3_ and microelements (B, Cu, Fe, Mn, Mb, Zn) (Azofoska, INCO GROUP, JSCo, Warsaw, Poland) was applied. 

### 2.3. Analytical Methods

#### 2.3.1. Samples Extraction and Preparation

To recover petroleum products, zeolite samples after a particular experimental interval were extracted 3 times with dichloromethane (DCM, analytical grade) in an ultrasonic bath (T = 30 °C, 15 min. each time). All aliquots were pooled together, evaporated at room temperature and weighed to calculate percentage yield extracted. For each grease and oil extract, around 0.05 g was taken for gas chromatography-mass spectrometry (GC-MS) analysis. However, for heavy naphtha, which was mostly removed even at the earliest stage of the experiment, the whole extract was used. Extract samples for GC-MS were dissolved in 1.5 mL DCM and analyzed. The composition of untreated petroleum products was also investigated. Around 0.05 g of each petroleum product was dissolved in 1.5 mL DCM. Only heavy naphtha and oil dissolved completely in DCM medium, whereas grease dissolved partially. Solid, non-solute residue was filtrated and only the DCM-soluble fraction was analyzed on GC-MS.

#### 2.3.2. Gas Chromatography-Mass Spectrometry (GC-MS)

GC-MS was used to analyze the total extracts, since the sample composition was uniform and consists of only aliphatic hydrocarbons. All extracts were analyzed on an Agilent 7890A gas chromatographer with DB-5 column (60 m × 0.25 mm i.d.) coated with a 0.25μm stationary phase film coupled to a 5975C XL MDS mass spectrometer. The carrier gas was He. The temperature program was set as follows: 50 °C for 2 min, (2) 175 °C at 10 °C/min, (3) 225 °C at 6 °C/min, and finally, (4) 300 °C at 4 °C/min and held for 20 min. The spectrometer was operated in electron ionization (EI) mode (70 eV, full scan) and scanned for *m*/*z* ratio at 50–650 Da. The data was obtained in full-scan mode and processed with the Hewlett–Packard Chemstation software (Agilent MSD Chemstation Software G1701AA). Compounds were identified by their mass spectra (using peak areas acquired by the manual integration mode), by comparison of peak retention times with those of standard compounds, and by interpretation of MS fragmentation patterns and literature data (MDS Data, 2012, Philp 1985). Geochemical ratios were calculated using total ion chromatogram peaks integrated manually, or, for the same compound groups, e.g., *n*-alkanes, on the suitable ion chromatograms (*m*/*z* = 71). 

Geochemical ratios were selected to reflect media degradation dynamics as follows:(1)Ratio of two acyclic isoprenoids differing in a chain length by a CH_2_ group, i.e., pristane (C_19_), abbreviated as Pr, and phytane (C_20_), abbreviated as Ph;(2)Ratios of acyclic isoprenoids to *n*-alkane: pristane (Pr) to *n*-heptadecane (*n*-C_17_) and phytane (Ph) to *n*-octadecane (*n*-C_18_);(3)Ratios of *n*-octadecane (*n*-C_18_) to 17α(H), 21β(H)-hopane (abbreviated as H_30_) and *n*-undecane (*n*-C_11_) to 17α(H), 21β(H)-hopane (H_30_);(4)Ratios of two *n*-alkanes differing in a chain length, i.e., n-heptadecane (*n*-C_17_) to *n*-octadecane (*n*-C_18_), *n*-undecane (*n*-C_11_) to *n*-octadecane (*n*-C_18_), *n*-undecane (*n*-C_11_) to *n*-dodecane (*n*-C_12_), *n*-undecane (*n*-C_11_) to *n*-henicosane (*n*-C_21_), and n-henicosane (*n*-C_21_) to *n*-hentriacontane (*n*-C_31_). The *n*-alkane ratios were selected so that the whole range of n-alkanes was covered, i.e., from *n*-C_11_ to *n*-C_31_, to better follow the biodegradation dynamics.

All ratios were constructed in such a way that they decrease with increasing petroleum product degradation, i.e., content of the less resistant compound is divided by that of the more resistant one. Several ratios were required since the composition of petroleum products varies greatly. For example, heavy naphtha does not contain pentacyclic triterpenes (hopanes), a compound group relatively well resistant to biodegradation, whereas grease and oil do not contain lighter *n*-alkanes, so n-undecane ratios cannot be applied in these cases. 

#### 2.3.3. Electrical Conductivity (EC) and pH

The pH and EC were measured with a potentiometric method in a media suspension of distilled water ratio 1:5 (*v*/*v*). After each measurement, the electrode was washed of residual viscous fractions in acetone.

#### 2.3.4. Microorganisms

*Bacillus licheniformis* was used in the experiment as the effective species in the biodegradation of crude oil [58]. The pure colonies of *Bacillus licheniformis* NCBI 1402, which was used as an efficient PGA producer, were incubated in a broth culture medium (sodium glutamate 5 g, glucose 7 g, sodium citrate 0.7 g, ammonium sulfate 0.7 g, sodium nitrate 0.7 g, yeast extract 1.6 g, peptone 1.6 g, zinc sulfate 0.07 g, and dipotassium phosphate 0.07 g) for 48 h. A colony forming unit (CFU) of the samples was enumerated by a serial dilution technique: samples were serially diluted up to 10^8^ dilutions and plated into starch-mineral salt agar medium, composed of 10 g starch (soluble), 1.8 g K_2_HPO_4_, 4.0 g NH_4_Cl, 0.2 g MgSO_4_·7H_2_O, 0.1 g NaCl, 0.01 g FeSO_4_·7H_2_O, and 15 g agar in 1 L of distilled water. Incubation was carried out at 37 °C for 24 h, and after that CFU was calculated using manual colony counter (LMCC-A11) [60].

### 2.4. Statistical Analysis 

Since the hydrocarbon parameters were not normally distributed (determined by using the Shapiro–Wilk test), we used a generalized linear mixed model (GLMM) to check the relationship between each hydrocarbon and microbial parameter and PGA treatments and time. The pairwise comparison between treatments in given time period were performed using Duncan pairwise test (Statistica v. 13, TIBCO Software Inc., Palo Alto, CA, USA).

A redundancy analysis (RDA) was performed to explain the pattern of variability in hydrocarbon parameters and independent variables (PGA treatments and time). Both variables were applied in a 0–1 system as dummy variables. The significance of these explanatory variables was analyzed with forward selection of the RDA and a Monte Carlo test with 499 permutations. The RDA was performed with the Canoco software version 5 [61].

## 3. Results

### 3.1. Percentage Yield Extraction 

The percentage yield extraction gives information on how much of the hydrocarbons were recovered from the zeolite after the experiment and, indirectly, information on the hydrocarbon loss due to applied treatments. 

The generalized linear mixed model (GLMM) was used for heavy naphtha (N), oil (O), and grease (G) parameters, and treatment–time factors revealed that in the case of all tested petroleum products, the yield of the extract was significantly dependent on the combined effect of the applied treatment and the time factor. In the case of G, the effect of treatment and time as single factors was also significant, while in the case of N and O, time was important as an independent factor (Appendix A).

Initially, i.e., after 7 days from the beginning of the experiment, the highest yields of the extract were in the reference samples (PGA_0_) for all tested petroleum products, which may be related to the reduction of evaporation of light hydrocarbon fractions due to the use of PGA (Figure 1). In the days following, an increase in the yield of the extract was observed, especially for O and G. In this case, more extract from PGA_1B_ was obtained than from PGA_0_ treatment on day 28, but, as time passed, the yield for this treatment decreased, especially in O. For treatments with PGA_1_ and PGA_10_, extract yield increased from day 56, reaching a maximum on day 112, with no significant differences between the two PGA concentrations. For N, the maximum extract yield was on day 7 in PGA_0_, while in the following dates, the differences between treatments were not significant (Figure 1).

### 3.2. Hydrocarbon Mass Loss

The mass loss of the samples occurred both as a result of evaporation and degradation of certain hydrocarbon fractions, and the participation of each of these processes individually is difficult to estimate. Evaporation loss is a particularly significant factor in case of N (Figure 2) and it is attested by the reference samples since this petroleum product is composed mostly of lightweight compounds.

The results of the GLMM analysis showed that in the case of the mass loss of the samples, only single factors were of significant importance. For N and G, it was the use of PGA and time, while for O, only time was significant. No significant relationships were found for the combination of both factors for any of the petroleum products (Appendix A). The use of PGA in all variants of the experiment significantly differentiated the mass loss in relation to the reference (PGA_0_), and the greatest losses were found for the PGA_10_ treatment in N and O.

For all the petroleum products in PGA_0_, the hydrocarbon mass loss gradually increased over the entire experimental period. For various PGA treatments, an increase in mass loss in relation to PGA_0_ was observed after only 7 days from the beginning of the experiment. The maximum mass loss due to PGA application appeared on day 56, and then a decrease was observed, which was especially pronounced for O and G. This tendency is most likely related to the degradation of PGA and the slowing down of the degradation rate of hydrocarbons. In the case of N, it can be assumed that the evaporation of light-weight hydrocarbon fractions had a significant share in the mass loss already in the first days after the start of the experiment. For both O and G, there were significant differences in percentage mass loss in all PGA variants compared to the reference throughout the whole experiment. However, no significant differences were found between different PGA treatments.

### 3.3. Electrical Conductivity (EC) and pH 

When interpreting results, we had in mind that the measurement and analysis of electrical properties of grease (and other high-viscosity oil products which have a complex architecture) is difficult, especially when their main components are non-ionic hydrocarbons [62]. The use of the NPK (nitrogen-phosphorus-potassium) fertilizer influenced the pH values, and in particular the EC. However, the absolute values were not as important as the trend of the changes in these parameters in relation to the reference samples due to the use of PGA. 

The pH of the analyzed petroleum products was significantly influenced by both the treatment and time factors, as single factors and in combination (Appendix A). In general, N showed an increase in pH, while O and G showed a decrease in all PGA treatments relative to PGA_0_. The pH values for N and O differed significantly only in relation to PGA_0_, and the pH in both substances stabilized at a similar level. In the case of G, significant differences between the treatments were found, and the pH values were slightly higher. The lowest pH, close to neutral, was found in PGA_1_ (Figure 3).

The EC for N was significantly influenced by both treatment and time factors, also in combination. In turn, for O, only the application of PGA was important. For G, time was more important, also in combination with treatment (Appendix A). In case of N, the EC values remained below the value in PGA_0_ throughout the experiment, while for O and G remained above. This is related to the difficulties in measurement, since heavier hydrocarbons are poor conductors. In all products, however, the effect of PGA and significant differences between various variants of its use, especially in G and O, were observed. However, the lowest EC values (except PGA_0_) were found in the PGA_1B_ treatment in all tested products, decreasing in the order of N, O, and G (Figure 4).

### 3.4. Geochemical Ratios

GLMM analysis for N showed significant importance of the time factor in combination with PGA treatment for two geochemical ratios, i.e., *n*-C_11_/*n*-C_12_ and *n*-C_21_/*n*-C_31_. However, in the case of longer hydrocarbon chains (*n*-C_21_/*n*-C_31_), PGA treatment was not significant as a single factor (Appendix A, Appendix A). In the case of *n*-C_11_/*n*-C_12_ and *n*-C_21_/*n*-C_31_ ratios, the rate of the decomposition was similar. The highest values compared to PGA_0_ were observed on day 28 of the experiment for PGA_10_, while after day 56, the indices of all PGA treatments decreased to zero (Figure 5).

In case of O, for most geochemical ratios, (*n*-C_11_/*n*-C_18_, *n*-C_18_/H_30_, *n*-C_18_/Ph, *n*-C_17_/Pr) time, and treatment factors were individually significant. However, for *n*-C_11_/H_30_, only time and Pr/Ph treatment, and time in combination with treatment, were more important (Appendix A). The action of PGA for most ratios was revealed after 7 days. The effect of each treatment was particularly clear in case of *n*-C_18_/H_30_, *n*-C_18_/Ph, *n*-C_17_/Pr, where there was a significant difference in the index values in relation to PGA_0_. Interestingly, after day 56 in PGA_10_, the Pr/Ph value (and to a lesser extent also *n*-C_18_/H_30_ and *n*-C_18_/Ph) gradually increased up to day 112 (Figure 6). 

GLMM of G for the *n*-C_18_/H_30_ and Pr/Ph indices showed both treatment and time factors were significant (both individually and in combination). For *n*-C_17_/Pr, time was also important in combination with treatment, and for *n*-C_17_/*n*-C_18_, both factors were individually significant (Appendix A). However, no significant differences were found between the treatments for *n*-C_18_/Ph and *n*-C_17_/Pr. For most indices, a rapid decrease of the ratios was observed until day 28 of the experiment, after which it stabilized on a similar level for each treatment. However, the Pr/Ph ratio was the highest on day 56 and 112 of the experiment in both PGA_10_ and PGA_1_ (Figure 7). 

### 3.5. The Protective Role of PGA for Bacillus Strains 

The pattern of bacterial population growth differed with the petroleum products and the degradation intensity (Figure 8). There was a significant effect of PGA and PGA combined with time on the bacterial density during the degradation experiment (Appendix A). In N, the density of *Bacillus licheniformis* increased after 7 days of treatment (Figure 7). On day 28 of the experiment, there were still higher densities of bacterial strains in PGA treatment than in the reference one. The density of bacteria dropped to zero after day 112 of the experiment. In O and G, similar pattern of bacterial dynamics during the experiment was observed (Figure 7). Surprisingly, we observed the peak of the CFU after 56 days in O, and 28 days in G. 

### 3.6. Multivariate Analysis

The redundancy analysis (RDA) for N revealed that the first axis explained 93.2% and the second 5.9% of the variance of the analyzed factors describing degradation. The gradient of variation of the first axis was related to T7, while in the case of the second axis, it was related to PGA_0_ (Table 1). Variables such as pH, hydrocarbon mass loss, and *n*-C_11_/*n*-C_12_ ratio were positively correlated with treatment (PGA_1B_), while EC, extract yield, *n*-C_11_/*n*-C_21_ ratio were related to time (T7). The dependence of PGA_1B_ and PGA_10_ on hydrocarbon mass loss was positive, and negative with respect to the extract yield. The time gradient was more important for geochemical indices than for the rest of variables (pH, EC, hydrocarbon mass loss, and extract yield) (Figure 9). 

The RDA for O degradation parameters described 73.1% and 22.6% variance of the dependent-independent variable relation for the first two axes, respectively. For the first ordination axis, the gradient was mostly positively related to PGA_0_ and T28 and negatively to PGA_10_, PGA_1B_, and T7 (Table 1). There was a visible negative correlation between PGA_1_ and all geochemical ratios. Hydrocarbon mass loss was more strongly associated with the treatments PGA_1_ and PGA_10_ than with PGA_1B_, while the extract yield was not related to any of analyzed factors. As in the case of N, the effect of PGA_1_ was positively related to T28 and T56, while T7 of the experiment was the most significant for the PGA_1B_ treatment (Figure 9).

In case of G, the RDA showed a different pattern of variables distribution (Figure 9). The first two ordination axes described 86.5% (axis 1) and 12.9% (axis 2) of variance for decomposition variables and environmental factors’ relation. The first axis was mostly described by PGA application, while the second axis was related to the first period of the experiment (T7) (Table 1). All analyzed geochemical ratios, except for Pr/Ph, were positively correlated with T7 and negatively with PGA_1B_, and less positively correlated with PGA_10_. On the other hand, PGA_1_ was of greater importance for the Pr/Ph ratio. The increase of hydrocarbon mass loss and extract yield were associated with a lower concentration of PGA (PGA_1_ and PGA_1B_).

## 4. Discussion

The results of the experiment show the positive effect of PGA on the degradation of various PH fractions, and its protective effect enabling the survival of the bacterial population in conditions of strong contamination in all tested petroleum products, i.e., heavy naphtha, lubricating oil, and grease. An increase in PH removal was observed, which was visible with an increase in the extract yield and hydrocarbon mass loss, as well as the changes of geochemical ratios in all treatments of the PGA in the experiment (1% PGA, 10% PGA, 1% PGA with *Bacillus licheniformis*) compared to the reference samples. PGA impact was of greatest importance, until around day 60.

The mechanism of PGA decomposition properties can be related to the anionic homopoliamid structure of the molecule and its action as a surfactant, which leads to the formation of micelles and the gradual release of PH absorbed in the zeolite. Due to the molecular weight, biosurfactants are classified into low- and high-molecular weights, which differ in their mode of action. Low-molecular-mass biosurfactants are effective in reducing surface and interfacial tension, while high-molecular-weight surfactants are effective in stabilizing emulsions [63]. *Bacillus licheniformis* produce the high-molecular-weight biosurfactant varying from 100 up to 2000 kDa molecules with gamma side branches [38]. In the case of the PGA applied in the experiment, its molecular weight ranged between 3000–140 kDa (unpublished data), hence a wide range of actions of PGA as a surfactant are possible on various PH fractions. 

The experiment demonstrated that in general for the stimulation of hydrocarbon degradation processes, using a lower concentration of PGA resulted in better performance than when using higher concentrations. The increase in surfactant concentration causes the surface tension to fall to the critical point (the so-called critical micelle concentration, or CMC), when the surfactant can form micelles [63]. In the initial stage of our experiment, the extract yield of different PGA treatments (concentrations) was lower than in the reference, which may be related to the formation of micelles under the influence of PGA. The mechanism of the formation of micelles from the PGA biopolymer, the hydrophobic elements of which are located in the micelle core and hydrophilic elements in the external environment, is presented in detail in [64]. Based on results of the experiment, we assume that hydrophobic, high-molecular-weight hydrocarbons are bound at the core of the micelles, while low-molecular-weight hydrocarbons remain outside. This is confirmed by the differences in the yield of heavy naphtha, lubricating oil, or grease extract. The yield of naphtha extract, where low-molecular PH fractions predominate (as opposed to oil and grease), equalize with the reference level after only 7 days. Conversely, the heavy hydrocarbons predominant in oil and grease are released from the micelle core as PGA decomposes after about 60 days, which is also indicated by a decrease in mass loss after day 56 of the experiment. The decomposition of PH continues to day 112, which is evident due to the increase in the extract yield in the experiment treatments with 1% and 10% PGA. This in turn indicates a gradual release of PH from the zeolite. In addition, the formation of micelles increases the contact surface of PH with oxygen, which may also accelerate their degradation [13].

The removal of the tested petroleum products was also reflected by changes in values of geochemical ratios, which indicated that the highest rate of degradation was at the first stage of the process. Since Pr is less resistant to biodegradation than Ph, *n*-C_18_ less than H_30_, and *n*-C_18_ less than Ph, such changes can be explained by the gradual release of less-degraded petroleum products from the PGA micelle. In the case of heavy naphtha, most of the compounds were removed at the first stages, within 28 days of the experiment, since it is testified by extract yield. Evaporation played a very significant role in the process here, despite zeolite application to retain hydrocarbons, as it is applied in soils and sediments. In the reference treatment, on day 28, only one of the five repetition samples contained any *n*-C_11_ detected by GC-MS analysis. After this stage, the remaining residue was composed of much heavier-weight compounds, and naphtha degradation dynamics at later stages were similar to petroleum heavy fractions, such as grease and oil. 

In the case of heavy naphtha, all geochemical ratios, as well as pH, showed the same statistically valid dependence on time, treatment, and time versus treatment (Appendix A, Appendix A). This is possibly because the compounds used here all belong to the same compound group, i.e., *n*-alkanes. However, much fewer uniform results were found for lubricating oil and grease. In the case of ratios composed only from *n*-alkanes, time and treatment showed a significant impact in both these media. The differences were found for ratios of *n*-alkanes for H_30_, possibly due to differences in PGA or bacteria access to these compounds caused by their different viscosities.

In the 1% PGA treatment with bacteria strain (PGA_1B_), the maximum yield of the extract occurred around day 30, after which the yield gradually decreased as the bacteria gradually used the easily available carbon from light PH fractions as a source of carbon and energy [2]. This coincides with the maximum growth of the bacterial population in oil and grease at about the same time. The main problem in the biodegradation of heavy hydrocarbon fractions is the inhibition of bacterial growth due to the lack of their availability in the liquid phase due to their poor solubility [54]. Thus, to access them as food, microorganisms have evolved several strategies. Some of the bacteria excrete biosurfactants that emulsify hydrocarbons, use enzymes reacting with hydrocarbons, or biopolymers that convert hydrocarbons into polar compounds that have better solubility in water. Only a few bacteria can utilize hydrocarbons directly [65,66,67,68]. To overcome the problem of hydrophobicity in high-weight PH fractions, which limits the transfer to microbial cells, cultivation of surfactant-producing bacteria or addition of a biosurfactant to the contaminated substrate should be added [69]. 

In the variant of the experiment 1% PGA with *Bacillus licheniformis*, it was shown that PGA protects bacteria against extinction under high concentration conditions of all tested petroleum products. Most microorganisms show the potential for soil remediation when the concentration of petroleum pollutants is less than 5%, and, after exceeding this level, this potential decreases [16]. Under the conditions of the laboratory experiment, the level of contamination was high, and the population of *Bacillus licheniformis* in combination with 1% PGA began to decrease in grease from around day 30, and in oil from around day 60, which may be related to the approximate time of PGA decomposition [70]. Poor survival of bacteria after introduction into the soil environment is the basic problem of any bioaugmentation process [71]. Our preliminary research results suggest that PGA has significant potential for use in this process. The high survival rate of bacteria with the addition of PGA, even at high hydrocarbon concentrations, was also evidenced in [55]. 

In natural conditions, microorganisms exist in an assemblage embedded in extracellular polymeric substances produced by themselves, which is termed as a biofilm [37]. It has been proven that PGA is an important compound of biofilm by some *Bacillus* strains [72]. The physiological function of PGA has not been fully investigated, but it is known that it can protect microorganisms from adverse environmental conditions by chelating toxic compounds such as heavy metals [49,50], or protect cells from fungal infections and prevent antibodies from accessing bacteria [73]. The experimental results suggest that the PGA may also have a positive effect on the reduction of the toxicity of PH in soil microorganisms. 

With a high concentration of PH, the drop in the biodegradation may result from nutrient limitation for microorganisms due to an increased number of heavy-weight hydrocarbons [69]. The petroleum hydrocarbons contain a lot of carbon, but they are very poor in other basic nutrients for microorganisms, especially nitrogen and phosphorus, which are needed for cellular metabolism [74], so they must be supplied with them. In the experiment, changes in the electrical conductivity (EC) of the tested petroleum products were related to the application of the NPK nutrients which were metabolized by bacteria. In the reference treatment (PGA_0_) much higher EC in naphtha than in oil and grease resulted from differences in the density of individual petroleum products. In substances of higher densities (oil, grease), the possibility of releasing ions from the nutrient medium and conducting electricity was lower [62]. However, after one week of the experiment, the EC stabilized to some extent in all tested petroleum products, which may be related to the saturation of free PGA polymer bonds with ions from the nutrient medium. Among the three variants of the experiment with PGA, the lowest mean EC values were observed in the variant with bacteria (PGA_1B_). In this case, the EC achieved lower and lower values in naphtha, oil, and grease, respectively. The greater the proportion of heavy-weight hydrocarbons in the tested petroleum product, the greater the decrease in EC was observed. This is due to the availability of carbon as an energy source for bacteria. In the case of naphtha, the availability of carbon was greater than in case of oil and grease due to the advantage of light-weight, easily biodegradable hydrocarbons. On the other hand, the lowest EC values in oil and grease resulted from the intensive use of the nutrient medium by bacteria, which were less able to break down heavy-weight hydrocarbons. 

A tendency to stabilize the reaction at the level of slightly acidic to neutral as a result of PGA application was observed in the experiment. The pH changed by PGA by about 0.45 units for naphtha and grease, and by about 0.67 units for oil. The acidic reaction of naphtha increased, and the slightly alkaline reaction of oil and grease was lowered. For each of the tested petroleum products, the smallest changes in pH in relation to the reference were recorded in the variant of the PGA experiment with bacteria. The changes in pH observed in the experiment suggest that application of PGA, which acted as a buffer, may have a positive effect on the biodegradation process. In remediation of soils contaminated with petroleum products with the use of microorganisms, changes in pH are observed [14]. Bioremediation tends to occur more frequently at pH levels near 7, and it is successful at pH values ranging from 6 to 8 [75,76].

## 5. Conclusions

The results of the laboratory experiment show the positive effect of PGA on the degradation of various PH fractions in the tested petroleum products i.e., heavy naphtha, lubricating oil, and grease. In all variants of PGA application, which were 1% PGA, 10% PGA, and 1% PGA with B. licheniformis, an increase in PH removal was observed. It was evidenced by an increase in the percentage of extract yield and hydrocarbon mass loss, as well as the changes in geochemical ratios, especially in case of aliphatic particles. However, for different parameters, time factor was also important. The anionic homopoliamid structure of the PGA molecule is responsible for the mechanism of PGA decomposition properties. Its action as a surfactant results in the formation of micelles and the gradual release of the high molecular weight hydrocarbons from the micelle core. Additionally, our preliminary results suggest that PGA has significant potential for use in the bioaugmentation process. Survival of *B. licheniformis* protected by application of 1% PGA under high concentration of all tested petroleum products turned out to be successful. In addition, a tendency to stabilize pH at the level optimum for the bioremediation process after PGA application is promising. In general, PGA impact was of greatest importance until around day 60. We need a more detailed survey on efficiency of PGA application in natural conditions, including the temperature range and the physical and chemical parameters optimal for natural soil microbiome.

## Figures and Tables

**Figure 1 ijerph-19-15066-f001:**
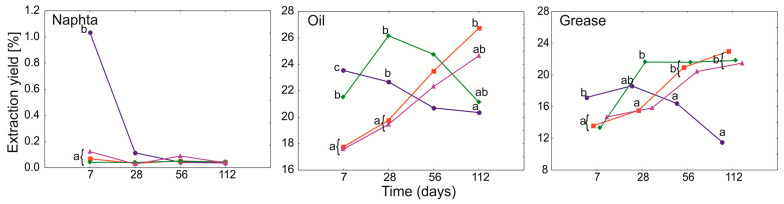
GLMM for mean extract yield of heavy naphtha, lubrication oil, and grease for PGA treatment (blue and circle—PGA_0_, red and square—PGA_1_, green and diamond—PGA_1B_, violet and triangle—PGA_10_) and time (days). The different letters indicate the pairwise differences according to Duncan pairwise test. Mean values, standard error and confidence intervals are included in Appendix A.

**Figure 2 ijerph-19-15066-f002:**
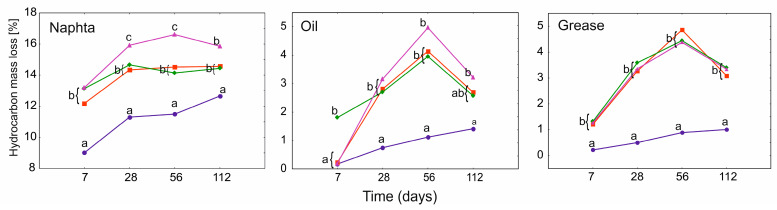
GLMM for mean hydrocarbon mass loss of heavy naphtha, lubrication oil, and grease for PGA treatment (blue and circle—PGA_0_, red and square—PGA_1_, green and diamond—PGA_1B_, violet and triangle—PGA_10_) and time (days). The different letters indicate the pairwise differences according to Duncan pairwise test. Mean values, standard error, and confidence intervals are included in Appendix A.

**Figure 3 ijerph-19-15066-f003:**
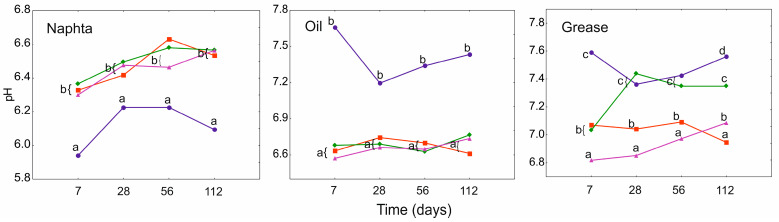
GLMM for mean pH (calculated from concentration of H^+^ ions) of heavy naphtha, lubrication oil, and grease for PGA treatment (blue and circle—PGA_0_, red and square—PGA_1_, green and diamond—PGA_1B_, violet and triangle—PGA_10_) and time (days). The different letters indicate the pairwise differences according to Duncan pairwise test. Mean values, standard error, and confidence intervals are included in Appendix A.

**Figure 4 ijerph-19-15066-f004:**
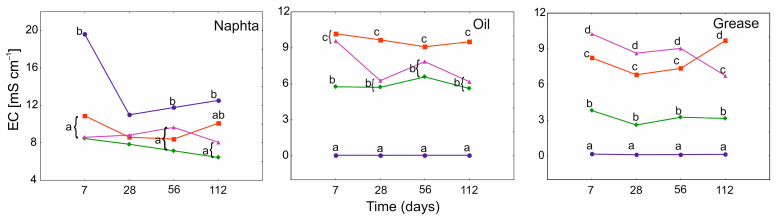
GLMM for mean electrical conductivity (EC) of heavy naphtha, lubrication oil, and grease for PGA treatment (blue and circle-PGA_0_, red and square-PGA_1_, green and diamond-PGA_1B_, violet and triangle-PGA_10_) and time (days). The different letters indicate the pairwise differences according to Duncan pairwise test. Mean values, standard error and confidence intervals are included in Appendix A.

**Figure 5 ijerph-19-15066-f005:**
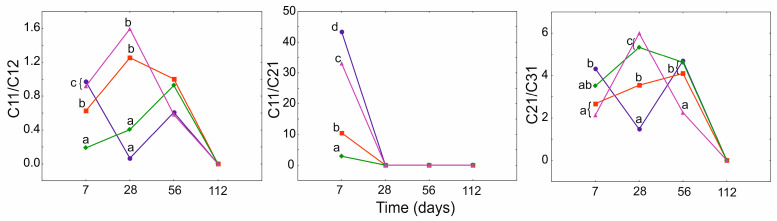
GLMM for geochemical ratios of heavy naphtha for PGA treatment (blue and circle—PGA_0_, red and square—PGA_1_, green and diamond—PGA_1B_, violet and triangle—PGA_10_) and time (days). The different letters indicate the pairwise differences according to Duncan pairwise test. Mean values, standard error and confidence intervals are included in Appendix A.

**Figure 6 ijerph-19-15066-f006:**
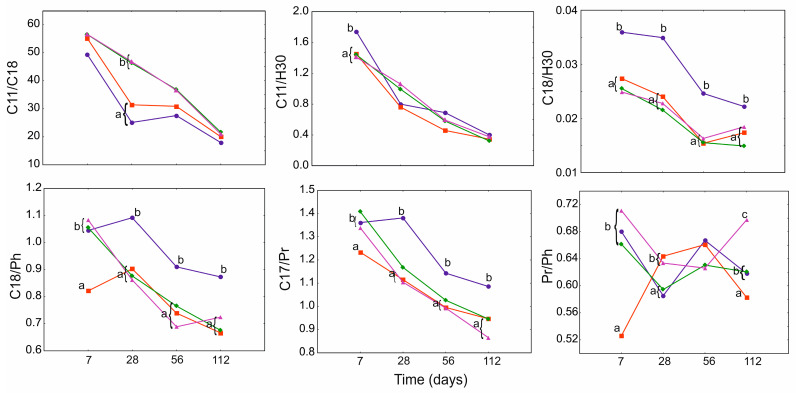
GLMM for geochemical ratios of lubricating oil for PGA treatment (blue and circle—PGA_0_, red and square—PGA_1_, green and diamond—PGA_1B_, violet and triangle—PGA_10_) and time (days). The different letters indicate the pairwise differences according to Duncan pairwise test. Mean values, standard error and confidence intervals are included in Appendix A.

**Figure 7 ijerph-19-15066-f007:**
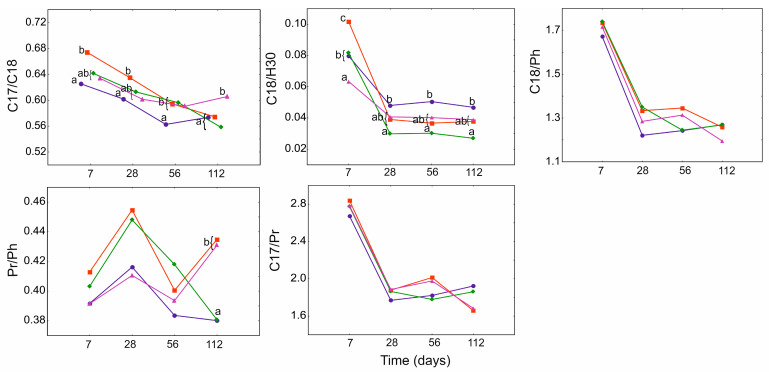
GLMM for geochemical ratios of grease for PGA treatment (blue and circle—PGA_0_, red and square—PGA_1_, green and diamond—PGA_1B_, violet and triangle—PGA_10_) and time (days). The different letters indicate the pairwise differences according to Duncan pairwise test. Mean values, standard error, and confidence intervals are included in Appendix A.

**Figure 8 ijerph-19-15066-f008:**
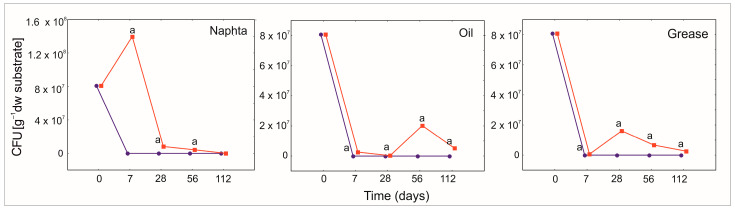
GLMM for microbial colony forming units (CFU) of heavy naphtha, lubrication oil, and grease for PGA_1B_ (red and square) and reference (blue and circle) treatment and time (days). The different letters indicate the pairwise differences according to Duncan pairwise test. Mean values, standard error, and confidence intervals are included in Appendix A.

**Figure 9 ijerph-19-15066-f009:**
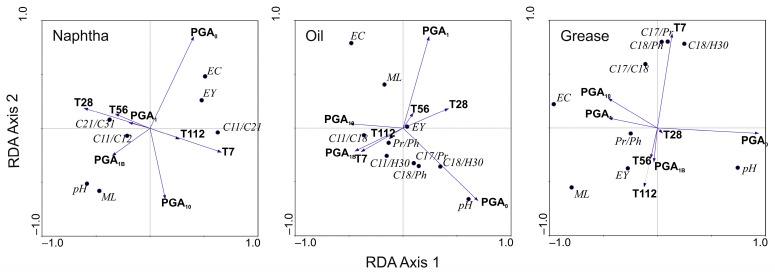
RDA for PGA treatments and time dummy variables and dependent variables for heavy naphtha, lubrication oil, and grease. PGA treatments: PGA_0_—reference, PGA_1_—1% PGA, PGA_1B_—1% PGA with *Bacillus licheniformis*, PGA_10_—10% PGA, Time: T7—day 7, T28—day 28, T56—day 56, T112—day 112. EY: extract yield. ML: hydrocarbon mass loss. EC: electrical conductivity. Geochemical ratios: *n*-C_11_/*n*-C_12_, *n*-C_11_/*n*-C_18_, *n*-C_11_/*n*-C_21_, *n*-C_17_/*n*-C_18_, *n*-C_21_/*n*-C_31_, *n*-C_11_/H_30_, *n*-C_17_/Pr, *n*-C_18_/Ph, *n*-C_18_/H_30_, Pr/Ph.

**Table 1 ijerph-19-15066-t001:** Inter set correlations of environmental dummy variables with two first redundancy axes for heavy naphtha, lubricating oil, and grease. PGA treatments: PGA_0_—reference; PGA_1_—1% PGA, PGA_1B_—1% PGA with *Bacillus licheniformis*, PGA_10_—10% PGA. Time: T7—day 7, T28—day 28, T56—day 56, T112—day 112.

Parameters	Heavy Naphtha		Lubricating Oil		Grease	
	AX1	AX2	AX1	AX2	AX1	AX2
PGA_0_	0.293	0.595	**1.203**	**−1.168**	**0.935**	0.034
PGA_1_	−0.142	0.039	0.419	**1.471**	−0.432	0.151
PGA_1B_	−0.253	−0.177	**−0.771**	−0.371	−0.026	−0.360
PGA_10_	0.102	−0.457	**−0.850**	0.069	−0.478	0.175
T_7_	**0.623**	−0.201	**−0.672**	−0.379	0.084	**0.841**
T_28_	−0.469	0.139	**0.730**	0.314	0.023	−0.171
T_56_	−0.234	0.093	0.164	0.239	−0.080	−0.440
T_112_	0.448	−0.158	−0.223	−0.175	−0.027	−0.230

## Data Availability

Not applicable.

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
