# Peer review of "Hydrocarbon Degradation and Microbial Survival Improvement in Response to γ-Polyglutamic Acid Application"

_ijerph, 2022, doi:10.3390/ijerph192215066_

Round 1

Reviewer 1 Report (New Reviewer)

The work of the authors on the study of the effect of PGA on the decomposition of hydrocarbons is highly relevant, especially for regions where the average annual temperature is low.

The methodological part is well presented and detailed. The authors carried out the necessary analytical work and statistical analysis. All important results are well presented in figures and tables, as well as in the supplemented materials. The list of references contains modern publications.

Some minor issues are:

1. Did the authors consider the possibility of optimizing the PH degradation process using increased air access, for example, periodic mixing?

2. Is there any data on the degradation process at different temperatures. The critical temperature for European ecosystems can be +15 C, when degradation processes slow down and any optimization can be very important.

Author Response

Reviwer 1

Open Review

English language and style

( ) Extensive editing of English language and style required
( ) Moderate English changes required
(x) English language and style are fine/minor spell check required
( ) I don't feel qualified to judge about the English language and style

Yes

Can be improved

Must be improved

Not applicable

Does the introduction provide sufficient background and include all relevant references?

(x)

( )

( )

( )

Are all the cited references relevant to the research?

(x)

( )

( )

( )

Is the research design appropriate?

( )

(x)

( )

( )

Are the methods adequately described?

(x)

( )

( )

( )

Are the results clearly presented?

(x)

( )

( )

( )

Are the conclusions supported by the results?

( )

(x)

( )

( )

Comments and Suggestions for Authors

The work of the authors on the study of the effect of PGA on the decomposition of hydrocarbons is highly relevant, especially for regions where the average annual temperature is low.

The methodological part is well presented and detailed. The authors carried out the necessary analytical work and statistical analysis. All important results are well presented in figures and tables, as well as in the supplemented materials. The list of references contains modern publications.

Some minor issues are:

  1. Did the authors consider the possibility of optimizing the PH degradation process using increased air access, for example, periodic mixing?

In the preliminary part of our reaserch on PGA application in bioremediation we wated to underestand the mechanizm of PGA action against PH. We decided not to introduce any additional factors at this stage to evoid difficulties interpreting results (what effect results from which facto at what level).

  1. Is there any data on the degradation process at different temperatures. The critical temperature for European ecosystems can be +15 C, when degradation processes slow down and any optimization can be very important.

We performer our experiment at constant temperature of 25°C, which is optimum for bioremediation.

We plan further studies using PGA to optimize other bioremediation conditions, like aeration.

Reviewer 2 Report (New Reviewer)

This paper presented a well-established experiment to investigate the function of poly γ-glutamic acid on lipid degradation and cytoprotection and showed that poly γ-glutamic acid promoted the degradation of petroleum hydrocarbons and was protective of bacterial population during bioremediation. In general, the English grammar is acceptable, but there are some elements that need to be corrected. Other specific comments are given below:

1.       The abstract should be more elaborative.

2.       Please check the name of the microorganisms, they must be written in Italic (such as in lines 16, 21).

3.       The sentence needs to be amended (line 132).

4.       Please note the abbreviations (such as CFU, B. subtilis)

5.       Please check the format of the References.

Author Response

Reviewer 2

Open Review

English language and style

( ) Extensive editing of English language and style required
( ) Moderate English changes required
(x) English language and style are fine/minor spell check required
( ) I don't feel qualified to judge about the English language and style

Yes

Can be improved

Must be improved

Not applicable

Does the introduction provide sufficient background and include all relevant references?

( )

(x)

( )

( )

Are all the cited references relevant to the research?

( )

(x)

( )

( )

Is the research design appropriate?

(x)

( )

( )

( )

Are the methods adequately described?

(x)

( )

( )

( )

Are the results clearly presented?

(x)

( )

( )

( )

Are the conclusions supported by the results?

(x)

( )

( )

( )

Comments and Suggestions for Authors

This paper presented a well-established experiment to investigate the function of poly γ-glutamic acid on lipid degradation and cytoprotection and showed that poly γ-glutamic acid promoted the degradation of petroleum hydrocarbons and was protective of bacterial population during bioremediation. In general, the English grammar is acceptable, but there are some elements that need to be corrected. Other specific comments are given below:

  1. The abstract should be more elaborative.

Abstract has been reconstructed.

Abstract: To improve the environmental sustainability of cleanup activities of contaminated sites there is a need to develop technologies that minimize soil and habitat disturbances. Cleanup technologies, like bioremediation, based on biological products and processes are in high future perspective. We studied the potential of γ-poly glutamic acid (PGA), as a natural component of biofilm produced by Bacillus sp., to be used for the decomposition of petroleum products, such as heavy naphtha (N), lubricating oil (O) and grease (G). The  study aimed to assess the impact of the use of different concentrations of PGA on the degradation process of various fractions of petroleum hydrocarbons (PH) and its effect on bacterial population growth in harsh conditions of PH contamination. In laboratory conditions, four treatments of PGA with each of the petroleum products (N, O, G) were tested: PGA0 (reference), PGA1 (1% PGA), PGA1B (1% PGA with Bacillus licheniformis), and PGA10 (10% PGA). After 7, 28, 56, 112 days of the experimentation, the percentage yield extraction, hydrocarbon mass loss, geochemical ratios, pH, electrical conductivity and microorganisms survival were determined. We observed an increase in PH removal, reflected as a higher amount of extraction yield (growing with time and reaching about 11% in G) and loss of hydrocarbon mass (about 4% in O and G) in all treatments of the PGA compared to the reference. The positive degradation impact was intensive until around day 60. The PH removal stimulation by PGA was also reflected by changes in values of geochemical ratios, which indicated that the highest rate of degradation was at the initial stage of the process. In general for stimulation of PH removal using a lower (1%) concentration of PGA resulted in better performance than a higher (10%). The PH removal facilitated by PGA is related to the anionic homopoliamid structure of the molecule and its action as a surfactant, which leads to the formation of micelles and the gradual release of PH absorbed in the zeolite carrier. Moreover, the protective properties of PGA against the extinction of bac-teria under high concentrations of PH were identified. Generally, the γ-PGA biopolymer helps to de-grade the hydrocarbon pollutants and stabilize the environment suitable for microbial degraders development.

  1. Please check the name of the microorganisms, they must be written in Italic (such as in lines 16, 21).

Corrected

  1. The sentence needs to be amended (line 132).

In the PGA1B treatment, pure cultures of Bacillus licheniformis (NCBI1402) at concentration level of 1x106 per 1 ml of broth solution were applied.

  1. Please note the abbreviations (such as CFU, B. subtilis)

Corrected

  1. Please check the format of the References.

Corrected

Reviewer 3 Report (New Reviewer)

(1           Abstract: Please include some of the important quantitative results.

(2)           Line 16, 21, and throughout the manuscript: Scientific name should be italicized.

(3)           Problem statement need to be highlighted.

(4)           Please state the objectives of this research clearly.

(5)           Line 71: Please add one more recent reference related to biodegradation: “Evaluation of Biodegradability of Poly(Butylene succinate-co-butylene Adipate) on the Basis of Copolymer Composition Determined by Thermally Assisted Hydrolysis and Methylation-Gas Chromatography. https://doi.org/10.1080/1023666X.2012.638439”.

(6)           Line 70, 73, and throughout the manuscript: References numbering should be simplified “18-22”, instead of “18, 19, 20, 21, 22”.

(7)           Line 77: Gram positive should be written with capital letter “G”.

(8)           Line 98: Please check. “area” or “are”?

(9)           Line 104: Missing full stop. “…PH contamination.”

(10)        Line 132: Missing full stop. “…PGA solution.”

(11)        Line 152, 210, and throughout the manuscript: “n-alkanes” should be written with italicized “n”.

(12)        Line 175: Please rephrase “However, only heavy naphtha and oil dissolved totally, …”.

(13)         Line 222-223: Glucose, Sodium, Ammonium, etc. should be written with small letter.

(14)        Line 227: 7H20-please include the subscript for 2.

(15)        All figures are too small. Please replace with bigger and clearer figures. Please provide information for the bullets of round, square, and triangle. What does it refer to?

(16)        Please add the Conclusion.

<<END>>

Author Response

Date of this review

02 Nov 2022 01:40:33

Reviewer 3

Open Review

English language and style

( ) Extensive editing of English language and style required
(x) Moderate English changes required
( ) English language and style are fine/minor spell check required
( ) I don't feel qualified to judge about the English language and style

Yes

Can be improved

Must be improved

Not applicable

Does the introduction provide sufficient background and include all relevant references?

( )

(x)

( )

( )

Are all the cited references relevant to the research?

( )

(x)

( )

( )

Is the research design appropriate?

( )

(x)

( )

( )

Are the methods adequately described?

( )

(x)

( )

( )

Are the results clearly presented?

( )

( )

(x)

( )

Are the conclusions supported by the results?

( )

( )

(x)

( )

Comments and Suggestions for Authors

  • Abstract: Please include some of the important quantitative results.

Added

(2)           Line 16, 21, and throughout the manuscript: Scientific name should be italicized.

Corrected

(3)           Problem statement need to be highlighted.

We highlighted the problem of bioremediation in the sentence: General objective of the study, according to modern trend of green remediation, is to apply natural biopolymers which are easily biodegradable and have no negative environmental impact.

(4)           Please state the objectives of this research clearly.

(5)           Line 71: Please add one more recent reference related to biodegradation: “Evaluation of Biodegradability of Poly(Butylene succinate-co-butylene Adipate) on the Basis of Copolymer Composition Determined by Thermally Assisted Hydrolysis and Methylation-Gas Chromatography. https://doi.org/10.1080/1023666X.2012.638439”.

Added

(6)           Line 70, 73, and throughout the manuscript: References numbering should be simplified “18-22”, instead of “18, 19, 20, 21, 22”.

Done

(7)           Line 77: Gram positive should be written with capital letter “G”.

Done

(8)           Line 98: Please check. “area” or “are”?

Improved

(9)           Line 104: Missing full stop. “…PH contamination.”

Done

(10)        Line 132: Missing full stop. “…PGA solution.”

Improved

(11)        Line 152, 210, and throughout the manuscript: “n-alkanes” should be written with italicized “n”.

Included in the text

(12)        Line 175: Please rephrase “However, only heavy naphtha and oil dissolved totally, …”.

The sentence was rephrased: Only heavy naphtha and oil dissolved completely in DCM medium, whereas grease dis-solved partially. Solid, not soluted residue was filtrated and only the DCM-soluble fraction was analyzed on GC-MS.

(13)         Line 222-223: Glucose, Sodium, Ammonium, etc. should be written with small letter.

Improved

(14)        Line 227: 7H20-please include the subscript for 2.

Yes, it was changed.

(15)        All figures are too small. Please replace with bigger and clearer figures. Please provide information for the bullets of round, square, and triangle. What does it refer to?

All letters and lines were enlarged and the solution of the figures was improved. We also added the explanations of bullets in the figure descriptions..

(16)        Please add the Conclusion.

Conclusions section has been added.:

  1. Conclusions

The results of the laboratory experiment show the positive effect of PGA on the degradation of various PH fractions in the tested petroleum products i.e. heavy naphtha, lubricating oil, and grease. In all variants of PGA application, which were 1% PGA, 10% PGA, 1% PGA with B. licheniformis, an increase in PH removal was observed. It was evidenced by an increase in the percentage of extract yield and hydrocarbon mass loss, as well as the changes in geochemical ratios, especially in case of aliphatic particles. How-ever, for different parameters time factor was also important. The anionic homopoliamid structure of the PGA molecule is responsible for the mechanism of PGA decomposition properties. Its action as a surfactant results in the formation of micelles and the gradual release of the high molecular weight hydrocarbons from the micelle core. Additionally, our preliminary results suggest that PGA has significant potential for use in the bioaugmentation process. Survival of B. licheniformis protected by application of 1% PGA under high concentration of all tested petroleum products turned out to be successful. Also a tendency to stabilize pH at the level optimum for the bioremediation process after PGA application is promising. In general PGA impact was of greatest importance until around day 60. We need more detailed survey on efficiency of PGA application in natural conditions including the temperature range, soil physical and chemical parameters or natural soil microbiome).

This manuscript is a resubmission of an earlier submission. The following is a list of the peer review reports and author responses from that submission.

Round 1

Reviewer 1 Report

The manuscript “ Hydrocarbon Degradation and Microbial Survival Improvement in Response to γ-Polyglutamic Acid Application” is well written and presented a good quality of work. Few important corrections are require in my opinion, that are mentioned bellow.

1] Author must be clearly write about hypothesis of manuscript and frame the objective of the work.

2] Line number 314-315- Hydrophobic, high molecular weight hydrocarbons
are bound at the core of the micelles, while low molecular weight hydrocarbons remain
outside.,
please justified this statement so that your next line based on unpublished data can be explain the presented work properly.

3] Line number 294-299 The mechanism of PGA decomposition properties can be related to the anionic mopoliamid structure of the molecule and its action as a surfactant, which leads to the
formation of micelles and the gradual release of PH absorbed in the zeolite. Due to the
molecular weight, biosurfactants are classified into low and high molecular weights,
which differ in their mode of action. Low molecular mass biosurfactants are effective in
reducing surface and interfacial tension, while high molecular weight surfactants are ef
fective in stabilizing emulsions.
Authors should be revised and maintain the flow of language, add suitable recent citation that supports the line --- Due to the
molecular weight, biosurfactants are classified into low and high molecular weights,
which differ in their mode of action. Low molecular mass biosurfactants are effective in
reducing surface and interfacial tension, while high molecular weight surfactants are ef
fective in stabilizing emulsions.

Line-450 to 456 author must be add details about why 3.3% total N , 6.1% P2O5, and 17.1% K2O were used in this experimental design, please justified and add suitable reference this is ½ dose of recommended fertilizer dose? See this article... Also write about how bacterial growth maintain? What population of bacteria was used during your experimental setup and this maintain during experiment… see this article https://doi.org/10.1111/plb.12173

4] Authors should mention at least any suitable reference which support trial and error methods design and mentioned in 4.1. Experiment Set-up section which justified the set-up of experiment – and  types of treatment of the petroleum products, heavy naphtha (N), lubricating oil (O), and grease (G), please see https://doi.org/10.1016/j.envres.2022.113081.

5] Several typographical and grammatically error was observed in this manuscript, please remove carefully such as line number 513 -10−8, section 4.3.4. Microorganisms- author should mentioned reference for CFU calculation.

6] Re-structure conclusion section

Author Response

  1. Author must be clearly write about hypothesis of manuscript and frame the objective of the work.

A sentence was added to clarify the purpose of the study.

  1. Line number 314-315- Hydrophobic, high molecular weight hydrocarbons
    are bound at the core of the micelles, while low molecular weight hydrocarbons remain
    outside., please justified this statement so that your next line based on unpublished data can be explain the presented work properly.

To make it more clear, how we understand results of our PGA experiment we changed the sentence as follows: Based on results of the experiment we assume that hydrophobic, high molecular weight hydrocarbons are bound at the core of the micelles, while low molecular weight hydrocarbons remain outside.

  1. Line number 294-299 The mechanism of PGA decomposition properties can be related to the anionic homopoliamid structure of the molecule and its action as a surfactant, which leads to the formation of micelles and the gradual release of PH absorbed in the zeolite. Due to the
    molecular weight, biosurfactants are classified into low and high molecular weights,
    which differ in their mode of action. Low molecular mass biosurfactants are effective in
    reducing surface and interfacial tension, while high molecular weight surfactants are ef
    fective in stabilizing emulsions. Authors should be revised and maintain the flow of language, add suitable recent citation that supports the line --- Due to the
    molecular weight, biosurfactants are classified into low and high molecular weights,
    which differ in their mode of action. Low molecular mass biosurfactants are effective in
    reducing surface and interfacial tension, while high molecular weight surfactants are ef
    fective in stabilizing emulsions.

Citation have been added.

  1. Line-450 to 456 author must be add details about why 3.3% total N , 6.1% P2O5, and 17.1% K2O were used in this experimental design, please justified and add suitable reference this is ½ dose of recommended fertilizer dose? See this article... Also write about how bacterial growth maintain? What population of bacteria was used during your experimental setup and this maintain during experiment… see this article https://doi.org/10.1111/plb.12173

Using the fertilizer to support bacteria with nutrients was tested in preliminary laboratory studies on biodegradation of petroleum pollutants by using a biopreparation based on autochthonous bacteria with the addition of γ-PGA in the inoculation process. The results were published by Wojtowicz K., Steliga T., Kapusta P., Brzeszcz J., Skalski T. 2022. Evaluation of the Effectiveness of the Biopreparation in Combination with the Polymer γ-PGA for the Biodegradation of Petroleum Contaminants in Soil Materials 2022, 15, 400. https://doi.org/10.3390/ma15020400]. We added reference to subsection 2.1.

  1. Authors should mention at least any suitable reference which support trial and error methods design and mentioned in 4.1. Experiment Set-up section which justified the set-up of experiment – and types of treatment of the petroleum products, heavy naphtha (N), lubricating oil (O), and grease (G), please see https://doi.org/10.1016/j.envres.2022.113081.

The results of previously cited work were also used to design the experiment. We also cited the paper recommended by Reviewer.

  1. Several typographical and grammatically error was observed in this manuscript, please remove carefully such as line number 513 -10−8, section 4.3.4. Microorganisms- author should mentioned reference for CFU calculation.

Information on the method of CFU calculation has been added to subsection 2.3.4.

  1. Re-structure conclusion section.

We decided that the Conclusion section (not compulsory) is not necessary in case of our manuscript. The main conclusions of the results are summarized at the beginning of the subsequent paragraphs of Discussion section which an introduction to the detailed discussion of the results.

Reviewer 2 Report

Dear authors,

I believe that in the present design of the experiment you have not got clear answer for your question: influence of PGA on acceleration of HC degradation and on survival of Bacillus.  The variants of PGA0+ bacteria & and PGA10+ bacteria are missing.

Figure 8. It is shown that at time 0 number of bacteria in the reference was 8x10(7). How that could be because the reference variant did not contain bacteria [ 1) PGA0: zeolite + nutrients + petroleum product (reference)"]

Some conclusions in the Absract sounds questionable. Higher extraction yield in PGA variants does not mean increase of PH decomposition. Probably it should be more correct to use PH removal/release, The gradual release of PH does not mean "the hydrocarbon degradation abilities of PGA" - it is again facilitating just removal of HC. Please, check if term "probiotic" is appropriate to use in the context of your manuscript.

Methods. It is not clear how did you add bacteria without water? Description of this is missing. It is written that you counted bacteria on mineral medium - to grow bacteria the medium shoud contain also growth substrate. Why you took Bacillus strain NCBI - can it grow by HC or PH degradation?

All measurements were done in many replicates - it is worth to show this data with +/- deviation in Tables but probably better in Figures in Supplemental information.

Sentence " PH are mainly composed of different amounts of carbon and 54 hydrogen atoms, but they can also contain other heteroatoms such as nitrogen, oxygen, 55 and sulfur [2] ? is wrong. Hydrocarbons are compounds of carbon & hydrogen atoms. Compounds with heteroatoms are not hydrocarbons anymore.

Author Response

  1. I believe that in the present design of the experiment you have not got clear answer for your question: influence of PGA on acceleration of HC degradation and on survival of Bacillus.  The variants of PGA0+ bacteria & and PGA10+ bacteria are missing.
  2. Figure 8. It is shown that at time 0 number of bacteria in the reference was 8x10(7). How that could be because the reference variant did not contain bacteria [ 1) PGA0: zeolite + nutrients + petroleum product (reference)"]

Thank you very much for this remarks. It is clarified in subsection 2.1. In the beginning we used PGA0+bacteria treatment, however after 7 days all bacteria were extinct. We excluded this treatment from the analysis of hydrocarbon removal. We included information of this treatment only in subsection  3.5, figure 8. We also haven’t decided to add PGA10+bacteria variant, because our preliminary results showed that in the variant PGA1+bacteria the concentration of protective polymer is good enough to protect the strains.

  1. Some conclusions in the Abstract sounds questionable. Higher extraction yield in PGA variants does not mean increase of PH decomposition. Probably it should be more correct to use PH removal/release, The gradual release of PH does not mean "the hydrocarbon degradation abilities of PGA" - it is again facilitating just removal of HC. Please, check if term "probiotic" is appropriate to use in the context of your manuscript.

Yes, we fully agree with such interpretation of the processes and use of the expration „PH removal” insted of „PH decomposition/degradation”. We have made the appropriate corrections in the text.

The expression „probiotic properties” is changed to „protective properties”.

  1. Methods. It is not clear how did you add bacteria without water? Description of this is missing. It is written that you counted bacteria on mineral medium - to grow bacteria the medium shoud contain also growth substrate. Why you took Bacillus strain NCBI - can it grow by HC or PH degradation?

We have developed the bacteria application method in section 2.5. Bacteria were added with a broth (liquid medium). Yes, we didn’t place starch in the recipe of growing media by mistake. We added this information to the same subsection.

  1. All measurements were done in many replicates - it is worth to show this data with +/- deviation in Tables but probably better in Figures in Supplemental information.

We added extra table (Table S1) with this information to the Supplementary materials. We have used generalized linear modeling with Poisson distribution. In that case standard deviations would not be appropriate. Confidence interval values are included in the supplementary table.

  1. Sentence " PH are mainly composed of different amounts of carbon and 54 hydrogen atoms, but they can also contain other heteroatoms such as nitrogen, oxygen, 55 and sulfur [2]? is wrong. Hydrocarbons are compounds of carbon & hydrogen atoms. Compounds with heteroatoms are not hydrocarbons anymore.

Yes we agree, it was an obvious mistake. We removed this sentence because it was imprecisely translated, and at the same time it is not necessary.

Round 2

Reviewer 1 Report

Dear authors

Your revision is substandard, and you claimed that you had addressed every issue in the updated document. This is poor writing practise; we urge you to make meaningful and accordant changes.

1] Line-450 to 456 author must be add details about why 3.3% total N , 6.1% P2O5, and 17.1% K2O were used in this experimental design, please justified and add suitable reference this is ½ dose of recommended fertilizer dose? See this article... Also write about how bacterial growth maintain? What population of bacteria was used during your experimental setup and this maintain during experiment… see this article https://doi.org/10.1111/plb.12173

Your reply not justified the present research; Using the fertilizer to support bacteria with nutrients was tested in preliminary laboratory studies on biodegradation of petroleum pollutants by using a biopreparation based on autochthonous bacteria with the addition of γ-PGA in the inoculation process. The results were published by Wojtowicz K., Steliga T., Kapusta P., Brzeszcz J., Skalski T. 2022. Evaluation of the Effectiveness of the Biopreparation in Combination with the Polymer γ-PGA for the Biodegradation of Petroleum Contaminants in Soil Materials 2022, 15, 400. https://doi.org/10.3390/ma15020400]. We added reference to subsection 2.1. add suitable proper citation so that justify this study.

2] Authors should mention at least any suitable reference which support trial and error methods design and mentioned in 4.1. Experiment Set-up section which justified the set-up of experiment – and types of treatment of the petroleum products, heavy naphtha (N), lubricating oil (O), and grease (G), please see https://doi.org/10.1016/j.envres.2022.113081.

Authors stated that “The results of previously cited work were also used to design the experiment” this is not true and not mentioned in revised manuscript, please correct suitable justification and reference

3] Several typographical and grammatically error was observed in this manuscript, please remove carefully such as line number 513 -10−8, section 4.3.4. Microorganisms- author should mentioned reference for CFU calculation.

Authors stated that “Information on the method of CFU calculation has been added to subsection 2.3.4” , please add suitable justification.

Author Response

Dear Reviewer,

thank you very much for most of the suggestions. We believe that it can improve the quality of the paper. Some remarks are unclear for us, but we hope we explained all the issues. The numbering of lines and subsections used by the Reviewer does not agree with the submitted version of the manuscript, so if we have not made some corrections, please indicate them precisely.

Dear authors

Your revision is substandard, and you claimed that you had addressed every issue in the updated document. This is poor writing practise; we urge you to make meaningful and accordant changes.

1] Line-450 to 456 author must be add details about why 3.3%  (it should be 13,3%) total N , 6.1% P2O5, and 17.1% K2O were used in this experimental design, please justified and add suitable reference this is ½ dose of recommended fertilizer dose?

It is mentioned in line 162-165: “To obtain the good conditions for bacterial growth, the NPK fertilizer containing 13.3% total N (5.5% NO3, 7.9% NH4), 6.1% P2O5, 17.1% K2O and 4.5% MgO, 21.0 % SO3 and micro-elements (B, Cu, Fe, Mn, Mb, Zn) (Azofoska, Grupa INCO S.A.) was applied.” We have used commercially available fertilizer called Azofoska, where all components are balanced and controlled. We haven’t used any test to apply the fertilizer. Our team previous works confirmed the Azofoska usage as a good fertilizer for bacterial growth in hydrocarbon degradation experiments (http://dx.doi.org/10.1016/j.biortech.2012.08.092; DOI 10.1007/s11270-008-9971-x). In our experiments we adopted previous findings. We didn’t use any recommended doses of fertilizer as it is in case of soil. In our laboratory experiment doses of the fertilizer were calculated based on carbon content at tested products to achieve the proportion of C:N:P approx.100:10:1 (line 131) suggested in literature cited as optimum. For example, the density of heavy naphta is 0.75 g cm-3, so the content of carbon in hydrocarbons (excluding hydrogen) is about 31.25g in 50ml of the product used, so we need to add 3.125 g of nitrogen. Knowing that the total nitrogen content in Azofoska fertilizer is 13,3% we calculated the appropriate dose.

See this article... Also write about how bacterial growth maintain?

The bacteria have grown in each container where hydrocarbons were used as source of carbon, fertilizer as a source of nitrogen and macroelements. We did not use any other amendments (like water, aeration), because we wanted to observe specifically the effect of PGA on bacteria survival. It would be difficult to assess such effect in case of many factors involved in the process.

What population of bacteria was used during your experimental setup and this maintain during experiment… see this article https://doi.org/10.1111/plb.12173

We have mentioned it in the sentences (line 223-225): “Bacillus licheniformis was used in the experiment as the effective species in biodegradation of crude oil [57]. The pure colonies of Bacillus licheniformis (NCBI 1402) , which was used as an efficient PGA producer, were incubated in a broth culture medium (…)”  with the NCBI reference number of Maldi-tof identification.

Your reply not justified the present research; Using the fertilizer to support bacteria with nutrients was tested in preliminary laboratory studies on biodegradation of petroleum pollutants by using a biopreparation based on autochthonous bacteria with the addition of γ-PGA in the inoculation process. The results were published by Wojtowicz K., Steliga T., Kapusta P., Brzeszcz J., Skalski T. 2022. Evaluation of the Effectiveness of the Biopreparation in Combination with the Polymer γ-PGA for the Biodegradation of Petroleum Contaminants in Soil Materials 2022, 15, 400. https://doi.org/10.3390/ma15020400]. We added reference to subsection 2.1. add suitable proper citation so that justify this study.

We have cited the recent research where the same nutrition procedures were applied. The doses of the fertilizer were determined in laboratory experiments Water Air Soil Pollut (2009) 202:211–228, DOI 10.1007/s11270-008-9971-x and http://dx.doi.org/10.1016/j.biortech.2012.08.092.

2] Authors should mention at least any suitable reference which support trial and error methods design and mentioned in 4.1. Experiment Set-up section which justified the set-up of experiment – and types of treatment of the petroleum products, heavy naphtha (N), lubricating oil (O), and grease (G), please see https://doi.org/10.1016/j.envres.2022.113081. Authors stated that “The results of previously cited work were also used to design the experiment” this is not true and not mentioned in revised manuscript, please correct suitable justification and reference

We do not truly understand the reviewer suggestion that we didn’t use previously cited work to design the experiment. We haven’t repeated the scheme of the experiments but the results. We have applied the same fertilizer Azofoska, the proportion of C:N and we tested the protective role of PGA on bacteria on commercially used hydrocarbon products which are more toxic and cause environmental problems. The trial and error method was also applied in the cited article: Effectiveness of Bioremediation Processes of Hydrocarbon Pollutants in Weathered Drill Wastes, to determine dozes of fertilizers in lab experiments  (work by the members of the same research team as Doi: 10.3390/ma15020400). We suggest to add another citation: DOI 10.1007/s11270-008-9971-x.

3] Several typographical and grammatically error was observed in this manuscript, please remove carefully such as line number 513 -10−8, section 4.3.4.

Do you mean line 226 in subsection 2.3.4? It was previously corrected in resubmitted version. The concentration is 10-8 CFUml-1, but this is “up to 108 dilutions”. The other misspelling corrections are included in the text.

Microorganisms- author should mentioned reference for CFU calculation. Authors stated that “Information on the method of CFU calculation has been added to subsection 2.3.4” , please add suitable justification.

The CFU calculation method is so obvious that we thought that it is not necessary to put suitable justification. We added the reference to the CFU serial dilution–agar plate procedure. Capuccino, G. J., and N. Sherman. "Microbiology (a laboratory manual). The Benyamin." (2001).
